# One Health Approach to Tick and Tick-Borne Disease Surveillance in the United Kingdom

**DOI:** 10.3390/ijerph19105833

**Published:** 2022-05-11

**Authors:** Nicholas Johnson, Lawrence Paul Phipps, Kayleigh M. Hansford, Arran J. Folly, Anthony R. Fooks, Jolyon M. Medlock, Karen L. Mansfield

**Affiliations:** 1Vector-Borne Diseases, Virology Department, Animal and Plant Health Agency (APHA), Woodham Lane, Surrey KT15 3NB, UK; paul.phipps@apha.gov.uk (L.P.P.); arran.folly@apha.gov.uk (A.J.F.); karen.mansfield@apha.gov.uk (K.L.M.); 2Medical Entomology and Zoonoses Ecology, UK Health Security Agency, Porton Down SP4 0JG, UK; kayleigh.hansford@phe.gov.uk (K.M.H.); jolyon.medlock@phe.gov.uk (J.M.M.); 3International Development Team, Animal and Plant Health Agency (APHA), Woodham Lane, Surrey KT15 3NB, UK; tony.fooks@apha.gov.uk

**Keywords:** tick, surveillance, pathogens, citizen science, animal health

## Abstract

Where ticks are found, tick-borne diseases can present a threat to human and animal health. The aetiology of many of these important diseases, including Lyme disease, bovine babesiosis, tick-borne fever and louping ill, have been known for decades whilst others have only recently been documented in the United Kingdom (UK). Further threats such as the importation of exotic ticks through human activity or bird migration, combined with changes to either the habitat or climate could increase the risk of tick-borne disease persistence and transmission. Prevention of tick-borne diseases for the human population and animals (both livestock and companion) is dependent on a thorough understanding of where and when pathogen transmission occurs. This information can only be gained through surveillance that seeks to identify where tick populations are distributed, which pathogens are present within those populations, and the periods of the year when ticks are active. To achieve this, a variety of approaches can be applied to enhance knowledge utilising a diverse range of stakeholders (public health professionals and veterinarians through to citizen scientists). Without this information, the application of mitigation strategies to reduce pathogen transmission and impact is compromised and the ability to monitor the effects of climate change or landscape modification on the risk of tick-borne disease is more challenging. However, as with many public and animal health interventions, there needs to be a cost-benefit assessment on the most appropriate intervention applied. This review will assess the challenges of tick-borne diseases in the UK and argue for a cross-disciplinary approach to their surveillance and control.

## 1. Introduction

Disease surveillance is the systematic collection, analysis and dissemination of data on infections of importance to public or animal health. This informs appropriate action that can be taken to either prevent or limit further spread of disease. Data collected can then inform risk assessment, resource management and vaccination programmes where appropriate. For vector-borne diseases, the identification of the vector, its behaviour and abundance, and the infection rate of associated pathogens are also critically important to understand disease transmission dynamics. Ticks are one of the principal vectors responsible for the transmission of pathogens to humans [1], domestic animals [2] and livestock [3] throughout the world.

There is growing evidence that tick distribution and tick-borne disease transmission are increasing across Europe [4,5,6]. These trends are likely to be occurring in the United Kingdom (UK), although contemporary evidence for changes in disease incidence is limited. Climate change has resulted in milder winter temperatures at temperate latitudes, leading to an increase in tick survival and triggering tick activity earlier in the year. The most prevalent tick species in the British Isles is the common sheep or deer tick (*Ixodes ricinus*), which will seek hosts when temperatures rise above 7 °C for 5 days consecutively and when the saturation deficit is low [7]. These parameters determine the duration of periods during the year when ticks are active and when humans and livestock are at risk of exposure to tick-borne pathogens. Anecdotally, changes in activity as a result of climate change could be reflected in the earlier detection of tick-associated livestock infections in the UK and outbreaks of disease [8]. However, surveillance for the pathogens that cause tick-borne disease (see Table 1) in the vector has been limited in the UK, mostly focussing on *Borrelia burgdorferi* sensu lato [9], and often geographically restricted. There is also an urgent need to improve the scope of diagnostic pathogen testing for tick borne diseases that can support surveillance based on detection of disease. In addition to *I. ricinus*, which is particularly abundant in upland grazing areas and some lowland areas, the less abundant red sheep tick, *Haemaphysalis punctata*, has re-emerged as a source of infectious disease in livestock in southern England [10]. A similar trend has been observed for the ornate cattle tick, *Dermacentor reticulatus.* Beyond these dominant tick species, there is limited information on other ticks present in the UK and the pathogens they transmit [11]. Species such as the seabird tick, *Ixodes uriae*, are present in the UK and associated with many viruses, but rarely encounter mammalian hosts and do not appear to transmit diseases to them [12]. However, species such as *Ixodes hexagonus*, commonly associated with wild mammals, and *Ixodes frontalis,* which feeds preferentially on birds, can play a role in maintaining pathogens within wildlife reservoirs [13].

Two recent papers have comprehensively reviewed the threats of arthropod-borne pathogens posed to human [14] and animal [15] health in the UK by vectors such as ticks. The focus of this article is to outline how surveillance for ticks and tick-borne diseases can both enhance their control and mitigate their impact.

## 2. The Common Sheep Tick, *Ixodes ricinus*

While other Ixodid or hard tick species can be encountered within the UK, one species dominates in terms of distribution, abundance, biting and disease transmission (Figure 1A–D). *Ixodes ricinus*, is the most abundant and widespread tick species in mainland Europe [16]. Despite its widespread distribution, the species is highly susceptible to desiccation when away from a vertebrate host and typically favours areas with moderate to high levels of rainfall. As for most hard ticks, *I. ricinus* spends the vast majority of its life-cycle within the vegetative layer, surviving in areas where humidity levels remain above 80%. Across Europe, this species is found in deciduous woodland or mixed forest. In some countries, notably the UK, *I. ricinus* is abundant in upland grazing areas, as well as in lowland grazed grassland, including conservation grazing. It is also increasingly being discovered in urban green spaces such as public parks, where the opportunity to encounter and transmit pathogens to humans and companion animals may be significant [17]. Factors such as re-wilding, promotion of wildlife corridors and urban biodiversity have been suggested for increasing risk of encountering ticks [18].

*Ixodes ricinus* host seeking activity is thought to follow two temporal peaks, one during the spring and early summer between April and June, and then a second late summer peak between August and September, although there is little evidence in southern England for this second peak. As a consequence of diapause, all feeding stages of *I. ricinus* will have emerged during summer and autumn, including those ticks which had fed during summer/autumn of the previous year [19]. There is generally much reduced to no activity during the winter months where temperatures are too low. *Ixodes ricinus* is a generalist feeder and has been reported on over 160 vertebrate hosts [20], with adults showing a preference for large mammals including humans, sheep, cattle, and dogs. As a consequence, *I. ricinus* can act as a vector for a large number of pathogens to a wide variety of hosts. Despite being one of the most highly studied tick vectors of disease, many questions remain about its ecology and effective means to reliably predict and control pathogen transmission by this species [21].

## 3. The Challenge of Non-Native Tick Species

There are over 20 species of ticks considered to be indigenous to the UK (Table 2). Non-indigenous species detected within the country are considered exotic and if they become established would be termed invasive [22].

A key priority of tick surveillance is to identify the routes of introduction into the country, identify the species of discovered ticks rapidly and investigate to assess the potential animal and public health impact. If necessary, control measures to prevent establishment of a population may be required. Past experience in the UK has identified a number of potential routes of entry for the introduction of ticks. This includes the movement of companion dogs, highlighted by the appearance of a cluster of cases of canine babesiosis, a disease exotic to the UK, in southern England [23], the importation of reptiles [24] and ticks attached to people returning from abroad [25]. Understanding the route of entry when linked to human activities offers a means of prevention through public education and rapid tick identification when it does occur. An alternative means of entry is by attachment to migratory birds [26,27]. Whilst this route of introduction cannot be prevented it can explain the appearance of exotic ticks in the absence of human or animal movements. A recent example has been the detection of a male *Hyalomma rufipes,* a competent vector of Crimean Congo haemorrhagic fever virus (CCHFV), on a horse that had not travelled outside of the UK [28]. *Hyalomma rufipes* is indigenous in Africa and a possible explanation was the introduction of the tick as an engorged nymph attached to migrating swallows (*Hirundo rustica*) during the Northward spring migration from Africa along the Western Palearctic route. This was considered likely as swallows were nesting at the location where the horse was stabled.

With the exception of infestations of domestic properties by the dog tick, *Rhipicephalus sanguineus* [29] there is currently no evidence that an exotic tick species has established a breeding population within the UK. Whilst existing passive surveillance activities do detect incursions of non-native tick species in the UK, active surveillance is needed to detect and identify exotic ticks early and recognise the disease risks they present. Early detection offers the best opportunity to eliminate an invasive species before it becomes established. History is replete with examples of the translocation of ticks that have resulted in the emergence of devastating diseases of livestock including the introduction of heartwater from Africa into the Caribbean by *Amblyomma variegatum* ticks [30], Theileriosis into New Zealand by the Asian long-horned tick, *Haemphaphysalis longicornis* [31], and the potential for reintroduction of bovine babesiosis into the United States by *Rhipicephalus* spp. from Mexico [32]. Among one of the best examples is that of *Rhipicephalus microplus*, originally a resident of South-East Asia, but which has successfully established in many regions of Africa, Australia and South America [33] and is a threat to the livestock industry wherever it establishes.

On-going surveillance, in the form of submissions of ticks by members of the public and private veterinary surgeons (PVS), has been instrumental in identifying the species associated with these occurrences and identified the routes of entry. The source of an outbreak of canine babesiosis in southern England in 2016 was initially recognised by a PVS treating anaemic dogs and traced to a population of *Dermacentor reticulatus* ticks [23]. Isolated populations of this tick are found at coastal sites around England and Wales [34]. However, the focus for the disease in dogs was an urban location north of London, and field surveys detected *D. reticulatus* ticks infected with *B. canis* in grassy areas adjacent to a carpark where owners exercised their dogs. In the absence of this pathogen in the UK, it is likely that the ticks were recently introduced on dogs entering the country from southern Europe [35] or contact between an imported infected dog and resident uninfected ticks. A recent report has identified a male tortoise tick, *Amblyomma* species from a consignment of 800 leopard tortoises (*Stigmochelys pardalis*) imported from Zambia [36]. This follows an ongoing trend for tick importation into Europe on reptiles [37].

## 4. Surveillance for Ticks and Tick-Borne Disease

Surveillance aims to acquire information that can inform both disease prevention and control measures. The acquisition of information on ticks and tick-borne diseases can be facilitated by a range of methods.

The most commonly used method of surveillance is the monitoring of disease trends in ‘at risk’ populations. This is mainly reactive to clinical disease presentation and reliant on diagnostic confirmation of the disease in samples submitted from the patient or infected animal. Collation of data in time and space can identify trends in disease incidence and suggest corrective actions. A retrospective study of Lyme borreliosis in the UK between 1998 and 2016 demonstrated an increase in annual incidence [38]. This was attributed to a genuine increase in cases, increased awareness on the part of the public or general practitioners, or a combination of all three. It also highlighted a higher incidence of cases in patients’ resident in rural areas and in areas of lower economic means. Conversely, a survey of farmers and veterinarians in the Republic of Ireland suggested that the incidence of bovine babesiosis had decreased between the 1980s and 2013 [39]. Such activities provide a baseline from which future surveillance data can be compared, trends identified and mitigation, if required, can be applied.

There are a range of methods for both surveying and collecting ticks from the environment, the most appropriate being dependent on the behaviour of the tick and the resources available for the collection process [40]. Identifying areas to survey is usually directed by reports of tick biting from humans or on livestock, reports of disease or specific vegetation types that favour certain tick species.

For tick species that ambush potential prey whilst questing on the tips of vegetation, a common method for collecting ticks on vegetation is that of ‘dragging’ or ‘flagging’. This involves drawing a cloth or wool blanket attached to a pole over the vegetation. Questing ticks will attach to the cloth and can be collected. Pale material is favoured so that the ticks are easily visible against a light background. This can be used in an unstructured way to detect the presence of ticks and collect large numbers of all life stages of the resident tick population. Alternatively, structured surveys using standard sizes of cloth, are drawn at repeated intervals over a defined area to compare estimated density in different locations and at different times [41,42]. Other variables can be collected during field sampling, such as temperature and humidity, dominant vegetation types and evidence for key animal hosts that might be supporting local tick populations. This approach can be used as a direct indicator of public health risk, particularly when combined with local pathogen prevalence estimates. Although inexpensive in terms of equipment, flagging can be labour-intensive, requiring regular site visits and intensive collection of data during each sampling. Ticks collected from such field surveys can then be identified morphologically with the option to test for the presence of viral [43], bacterial [44] and protozoal [8] pathogens using a range of molecular methods. When applied consistently, and over a number of years, this approach can determine the prevalence and diversity of pathogens within a tick population or detect the emergence of a new pathogen within the population [45]. 

An alternative method to collecting ticks from the environment is the collection of ticks directly from animals, including humans. This gives the added benefit of providing some context to the tick-host-pathogen interaction, but also raises a number of problems. Clearly, removing a tick from an animal demonstrates that the tick is either feeding, or about to feed on a particular host. If attachment has occurred it can demonstrate the parts of the host to which the tick preferentially feeds such as the ears, axilla or inguinal areas [46,47]. Engorged ticks at all life stages are easier to spot compared to un-engorged counterparts, particularly with heavy infestations on livestock, although engorgement can make morphological identification more challenging as the tick can be swollen to many times its pre-feed size (Figure 1E). In addition, testing the engorged tick in its entirety for pathogens will not differentiate whether the tick is the source of the pathogen or the bloodmeal it has taken, or indeed whether the tick is capable of transmitting the pathogen. Removal of components of the tick such as the legs allows a level of discrimination between the feeding tick and the host bloodmeal, although this can be time consuming when dealing with large numbers of ticks. Some host species can be challenging to collect ticks from, with larger animals such as cattle and potentially dangerous animals requiring restraint or even sedation in order to visually inspect. Coverings of fur or feathers can further delay a thorough inspection of an animal. However, the benefits of establishing direct contact between the tick and the host have been used in many studies. For example, in Great Britain, collection of ticks from grouse chicks over two decades has provided strong evidence that *I. ricinus* ticks are increasing in abundance and distribution [48].

A growing trend in surveillance is to use non-specialists to contribute to data collection, so called “Citizen Science”. The traditional method to engage non-specialists to generate data on disease prevalence is through the use of surveys. Responses to a series of questions are analysed to identify a range of useful parameters associated with a particular problem. This remains a useful means of approaching tick-borne disease in both humans [49] and animals [38,50]. In the UK this has been augmented with a national Tick Surveillance Scheme delivered by the UK Health Security Agency [51,52] (formerly Public Health England). This scheme encourages members of the public to send ticks and related information, for identification and if novel, instigate further investigation. The data generated provides information on the distribution of different tick species across the UK, their seasonal activity and host associations. This not only identifies the key species of human and animal health importance, but also indicates the time of year when risk may be highest. Importantly, the scheme provides information back to anyone submitting samples, promotes One Health responses to unusual findings and disseminates this information across government departments. Across Europe, this approach has been used to map the presence and expansion of tick populations, identify hot-spots for tick-human interactions and generate samples for pathogen testing [53,54,55,56].

Targeted surveys have involved PVS submitting ticks that have resulted in publications on infestations in domestic dogs [57] and cats [58]. A recent approach has used electronic records submitted directly from veterinary surgeries to rapidly collate information on tick attachments and monitor trends in companion animals [59]. There have been no attempts to provide such surveillance for livestock. This would largely rely on those in farming communities or PVS in large animal practice to regularly provide basic surveillance data on tick encounters and detection of disease. Nevertheless, monitoring tick abundance in agricultural settings would provide a meaningful, cost-effective method for determining the distribution of ticks in rural areas and the likely impact of tick-borne disease.

Citizen science offers a number of benefits, especially the cost-effective generation of large datasets and engagement with non-specialists, e.g., the general public. However, it does have a number of limitations [60]. Firstly, it relies on citizens to be honest and accurate, with little opportunity to corroborate the accuracy of data submissions. Critical information, such as tick identification is usually beyond the ability of the general public and requires expertise in this area. It is no surprise that all the tick-associated projects cited above rely on submissions of ticks that are subsequently identified by experts. Another area of difficulty is establishing the geographical location where a tick first encounters a host and the location of the host when the tick is found. For humans this can, in extreme cases, be different continents due to international travel, although usually it relies on the submitter providing sufficient information to decide the most likely site of the original encounter, i.e., a recent outdoor excursion.

## 5. Detection of Tick-Borne Pathogens

The appearance of clinical signs in a susceptible host is usually the first step in detecting tick-borne disease. Some disease signs may provide a strong suspicion of a particular disease, for example the initial rash or erythema migrans that can sometimes develop in cases of Lyme borreliosis in humans and the appearance of blood in urine for bovine babesiosis. However, many disease signs are non-specific, such as febrile episodes, and require confirmation by a diagnostic test. This may involve isolation or microscopical identification of the causative agent, detection by serological means or molecular detection such as polymerase chain reaction assays.

For many pathogens of livestock, a large array of tests are now available to detect the aetiological cause of disease [61]. Serology can also be used to provide evidence of past exposure to tick-borne diseases. Ticks themselves can also be tested for the presence of pathogens [8] and pathogens are often reported in ticks before disease in hosts are reported. However, detection alone does not demonstrate causality without supporting evidence such as association with an infected animal. Indeed, novel microorganisms are regularly being detected in tick populations. Some are confirmed as pathogens affecting humans such as Severe Fever with Thrombocytopenia Syndrome virus [62]. However, many microorganisms have no apparent associations with disease and may not be capable of replicating outside of the tick host [63,64]. What is clear is that ticks are capable of hosting multiple viruses, bacteria and protozoa that can either individually or in combination cause disease in vertebrates [65].

## 6. A One Health Approach to Support Surveillance for Tick-Borne Diseases

The control of diseases has biological, social and political elements that require an interdisciplinary and collaborative approach in managing such health emergences [66,67]. The ‘One Health’ (1H) paradigm has been adopted as a tripartite initiative by the World Health Organisation (WHO), the Food and Agriculture Organisation of the United Nations (FAO) and the World Organisation for Animal Health (OIE). In a global health security context, the 1H concept recognises the complex interconnectedness and interdependence of humans, animals, and the environment, as well as the importance of dismantling disciplinary and professional silos. This approach involves multisectoral, interdisciplinary working on systems, strengthening efforts to prepare, detect, respond to, and recover from threats to human, animal and environmental health. Ticks transmit a range of pathogens that affect human and animal health, the biology of the tick is intimately linked to environmental and animal host factors, and tick-borne diseases can have a significant impact on public health and the livestock industry. A 1H approach is vital for ensuring effective and sustainable efforts to address control and prevention of tick-borne diseases.

## 7. Discussion

One tick species in the UK dominates both in terms of distribution and abundance, and in the transmission of endemic diseases. This is *I. ricinus*, a three-host tick that feeds on a diverse range of host species, including humans [52]. Other species of veterinary or public health significance are present in geographically restricted populations and are typically associated with pathogens of livestock or companion animals. The main focus of tick and tick-borne disease surveillance should be the understanding of the ecology and distribution of *I. ricinus* and its role in disease epidemiology. Nevertheless, the potential role of other UK tick species should always be considered.

The key objectives for UK surveillance for ticks and tick-borne pathogens should include:Long-term surveys to characterize the distribution and abundance of *I. ricinus* on a country-wide scale, and to better understand seasonality of activity and biting.Determination of the presence and prevalence of tick-borne pathogens in at risk host populations and within indigenous tick instars that transmit pathogens.Provide evidence-based information for public and animal health policy to identify, mitigate and control tick-borne pathogen transmission and disease spread.Rapid detection of introduced tick species and timely responses to reduce potential impact and prevent establishment or spread to the wider environment.Collate robust data and develop climate-based assessments for future exotic tick establishment and disease risk.

Tick identification is critically important to identify the species involved in human biting and to assess the clinical risk associated with a particular tick species [68]. For example, transmission of Lyme borreliosis occurs after attachment by infected *I. ricinus* adults and nymphs. For animal diseases, the detection of ticks within livestock habitats can confirm the source of disease. Identification of exotic ticks requires expertise to identify key morphological traits that should differentiate to genus level, but ideally to the species as this is critical to indicate potential pathogen associations, the populations at risk and implementation of control measures. If morphological identification is not possible either due to lack of expertise or access to detailed keys, or the state of the specimen precludes identification, other methods including polymerase chain reaction linked to sequencing or mass spectrometry [69] offer expensive and technically demanding alternatives, but which provide detailed evidence.

On discovery of ticks attached to humans, the tick should be removed in such a way as to completely remove the tick mouth parts from the wound to reduce the risk of pathogen transmission or development of a skin infection. For companion animals, ticks should be removed where possible and a spot-on acaricide treatment applied to prevent further infestations [70]. Livestock can also be treated with pour-on treatments, currently only licenced for tick treatment on sheep, although there is concern that this will accelerate acaricide resistance in *I. ricinus* that would reduce options for control.

Expert committees offer a critical role to assess disease risk, suggest intervention measures and identify research proposals. Within the UK, such committees include the Veterinary Risk Group [71] and the Human-Animal-Infections and Risk-Surveillance (HAIRS) group [72].

From a 1H perspective, national coordination and reporting of tick-borne diseases provides national level indicators to support an evidence-base for policy and advocacy to international organisation. This information is also relevant to community needs, priorities, and capacities, and should be included in national and community-based multisectoral coordination of tick-borne disease control. Citizen science projects can also be used to enhance surveillance projects by explicitly highlighting the role of the community. Going forward, surveillance in the UK should be enhanced by further developing a flexible and adaptable legal-policy framework with a national strategic plan and finance strategy for surveillance of tick and tick-borne diseases.

## 8. Conclusions

More focus is needed across the veterinary sector to match that of the public health sector, to quantify the true burden of TBD in animals within the UK, particularly monitoring changes to tick and TBD transmission affecting livestock, companion animals and humans. There is also a need to provide evidence for trends that might be driven by both climate and anthropogenic change. Critically this would include better integration and data sharing to support both public and animal health. A response along the lines outlined for American control of TBDs [73] is appropriate for the challenge faced by the UK. Coordinated investment is required to enhance vector-borne disease capability and provide capacity for the future, to develop competencies, testing and maintenance of expertise in support of veterinary efforts to control animal and zoonotic TBDs.

The benefits of well delivered surveillance include increased knowledge of tick distribution and abundance, and associated pathogens of public and animal health. This data can support disease vector distribution modelling and highlight risk areas for disease transmission when overlaid with other datasets such as susceptible livestock distribution and climatic factors that influence tick habitats and life-cycles. Comprehensive surveillance will also support studies that seek to understand genetic variation within tick and pathogen populations that might influence disease transmission and in the case of ticks, the genetic basis for the emergence of acaricide resistance. Finally, improved surveillance would provide the baseline for both vector and pathogen associations and incidence against which changes associated with climate change can be measured. Currently, change is being monitored by the appearance of disease in at risk populations, which is arguably too late to facilitate effective control measures.

## Figures and Tables

**Figure 1 ijerph-19-05833-f001:**
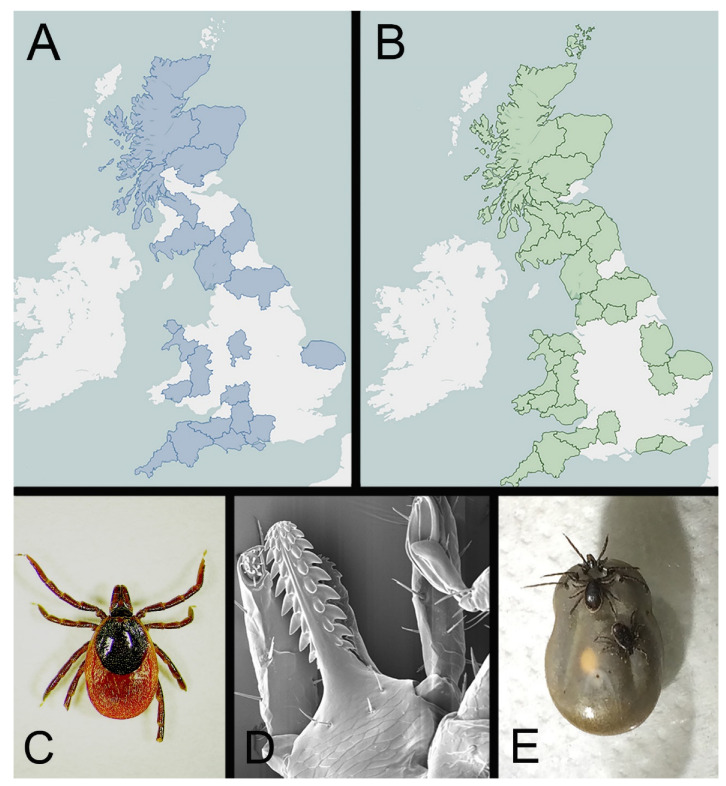
*Ixodes ricinus* and tick-borne disease within Great Britain. (**A**) Map of the British Isles showing the distribution (in blue) of cases of tick-borne disease in cattle (tick-borne fever, babesiosis and louping ill) reported by APHA between 2012 and 2021. (**B**) Map of the British Isles showing the distribution (in green) of cases of tick-borne disease in sheep (tick-borne fever, louping ill and tick pyaemia) reported by APHA between 2012 and 2021. (**C**) Image of adult female *Ixodes ricinus* (photo Arran Folly). (**D**) Scanning electron micrograph of the mouth parts of an *Ixodes ricinus* larva (photo Bill Cooley). (**E**) Image of an engorged female *Ixodes ricinus* with males (photo Nicholas Johnson). Maps generated by APHA’s Surveillance Intelligence Unit accessible at APHA Vet Gateway: Livestock disease surveillance dashboards (defra.gov.uk, accessed on 18 April 2022) using software under licence from https://public.tableau.com (accessed on 18 April 2022).

**Table 1 ijerph-19-05833-t001:** Key tick species and the tick-borne pathogens detected in the United Kingdom.

Tick Species	Habitat in the United Kingdom	Main Hosts	Pathogen	Disease
*Ixodes ricinus*	Deciduous and mixed forest, upland grazing areas, permanent lowland grazing areas	Immature forms feed on a variety of mammals, reptiles and birds. Adults favour large mammals including humans, livestock, domestic pets and deer	*Babesia divergens*	Bovine and human babesiosis
			*Babesia venatorum*	Human babesiosis
			Louping ill virus	Ovine encephalomyelitis
			*Borrelia burgdorferi* s.l.	Lyme disease
			*Borrelia miyamotoi*	Relapsing fever
			*Anaplasma phagocytophilum*	Tick-borne fever in ruminants
			Tick-borne encephalitis virus	Human encephalitis
			*Rickettsia* spp.	Disease not reported in UK
*Dermacentor reticulatus*	Coastal locations including sand dunes and grazing land	Adults feed on humans, livestock and domestic pets	*Babesia canis*	Canine babesiosis
			*Rickettsia* spp.	Disease not reported in UK
*Haemaphysalis punctata*	Chalk grassland and grazing marsh	Adults feed mainly on cattle and sheep. Occasional reports from dogs and humans	*Theileria luwenshuni*	Theileriosis in ruminants
			*Babesia motasi*	Ovine babesiosis
			*Babesia major*	Bovine babesiosis
			*Theileria orientalis*	Bovine theileriosis
			*Rickettsia massiliae*	Disease not reported in UK

**Table 2 ijerph-19-05833-t002:** The hard and soft tick species native to the United Kingdom.

Family	Species
Argasidae (Soft ticks)	*Argas reflexus*
	*Argas vespertilionis* *Ornithodoros maritimus*
Ixodidae (Hard ticks)	*Dermacentor reticulatus*
	*Haemaphysalis punctata*
	*Ixodes acuminatus*
	*Ixodes apronophorus*
	*Ixodes arboricola*
	*Ixodes caledonicus*
	*Ixodes canisuga*
	*Ixodes frontalis*
	*Ixodes hexagonus*
	*Ixodes lividus*
	*Ixodes ricinus*
	*Ixodes rothschildi*
	*Ixodes trianguliceps* *Ixodes univacatus*
	*Ixodes uriae* *Ixodes ventalloi* *Ixodes vespertilionis*

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
