# Peer review of "One Health Approach to Tick and Tick-Borne Disease Surveillance in the United Kingdom"

_ijerph, 2022, doi:10.3390/ijerph19105833_

Round 1
Reviewer 1 Report
Dear Authors,
The manuscript «One Health approach to tick and tick-borne disease surveillance in the United Kingdom» was interesting. There are only some minor changes, you should do.
L82: “ticks?”, the “?” why is there?
Figure 1: the caption should be under the figure
L298: “many” what?
Author Response
L82: “ticks?”, the “?” why is there?
Re. The question mark has been deleted.
Figure 1: the caption should be under the figure
Re. The figure caption has been moved.
L298: “many” what?
Re. “microorganisms” has been inserted into the text.
Reviewer 2 Report
This was a very well written review of ticks and tick-borne diseases in the UK. I only had a few very minor comments, listed below:
1) Line 86: I would delete "both" from this sentence
2) Line 163: "spp." should not be italicized
3) Line 170: Should this read "...canine babesiosis south of England in 2016..."
4) Line 336: Add a period to the end of the sentence
Author Response
1) Line 86: I would delete "both" from this sentence
Re. “both” has been deleted.
2) Line 163: "spp." should not be italicized
Re. “spp.” is now in normal text.
3) Line 170: Should this read "...canine babesiosis south of England in 2016..."
Re. The text has been edited to read “…canine babesiosis in southern England in 2016…”
4) Line 336: Add a period to the end of the sentence
Re. A period has been added.
Reviewer 3 Report
Review of a manuscript “One Health approach to tick and tick-borne disease surveillance in the United Kingdom” by Nicholas Johnson, L. Paul Phipps, Kayleigh M. Hansford, Arran J. Folly, Anthony R. Fooks , Jolyon M. Medlock , Karen L. Mansfield
Manuscript ID: ijerph-1711626
In the manuscript, the authors, based on the experience in the UK, indicate and describe the key goals of the surveillance of ticks and tick-borne diseases that will help both enhance their control and mitigate their impact. The authors interesting described strengths and weaknesses in the management and surveillance of tick-borne diseases in humans and animals in line with the 'One Health' paradigms that require an interdisciplinary and collaborative approach.
The text is well written and has a correct structure. The comments are detailed below.
Comments and suggestions:
p.2/Tab.1 - I suggest to complete the information on the main species of ticks in the UK by introducing data on the habitat and main hosts of these tick species
p.3/l. 95 - “… urban areas….” - please specify which habitats in cities are the most favorable for the existence of ticks
p.3/l. 99-100 - “…during the spring and early summer and then a second late summer peak…” - please indicate which months are the peak of the greatest activity of I. ricinus in the UK
p.4/Fig.1 A,B - maps a bit unreadable, I suggest introducing more contrasting colours for the areas of disease occurrence
p.4/Fig.1C,D,E - please insert name of the photos author
Author Response
p.2/Tab.1 - I suggest to complete the information on the main species of ticks in the UK by introducing data on the habitat and main hosts of these tick species
Re. Addition columns describing habitat and main host species have been added.
p.3/l. 95 - “… urban areas….” - please specify which habitats in cities are the most favorable for the existence of ticks
Re. The text has been modified to “..urban green spaces such as public parks, where…”
p.3/l. 99-100 - “…during the spring and early summer and then a second late summer peak…” - please indicate which months are the peak of the greatest activity of I. ricinus in the UK
Re. The text has been modified to “…early summer between April and June, and then a second late summer peak between August and September, although..”
p.4/Fig.1 A,B - maps a bit unreadable, I suggest introducing more contrasting colours for the areas of disease occurrence
Re. The colour selection is a default of the Tableau software and cannot be amended.
p.4/Fig.1C,D,E - please insert name of the photos author
Re. Photo authors have been added.